# BESTI: Bayesian Class-wise Trust-Weighted Ensemble with Structured Sampling for Imbalanced Multi-Class Classification

## Abstract

Class imbalance is caused by both practical and structural reasons, such as variance in occurrence frequency, biases in collection environments, differences in labeling costs, and imbalances in conceptual definitions. Such an imbalance introduces diverse problems, including under-representation of minority classes, distortion of metrics, and deterioration of model fairness and generalization capabilities. To challenge this issue, we propose a Bayesian Class-wise Trust-Weighted Ensemble with Structured Sampling for Imbalanced Multi-Class Classification, named BESTI. BESTI starts by constructing multiple sub-training sets from the original dataset that represent varying degrees of data imbalance. By initializing clipping thresholds in a structured manner, classes with larger sample counts than thresholds are downsampled, and others are retained. This creates a series of training sets, each reflecting a different class distribution. After, independent models are trained from each of the training sets, generating multiple specialized models for a certain degree of imbalance. We aggregate these models taking their trustworthiness into account. Based on Bayes' theorem, this trustworthiness is equivalent to the class-wise precision of the model. Utilizing this precision as a weight, BESTI ensembles multiple models to make the final decision. Our test results show that BESTI successfully improves the overall performance of the model, including the minority classes. In addition to that, BESTI shows competitiveness compared to state-of-the-art methods, often outperforming them significantly in certain domains.

## 1 Introduction

In the real-world, data is rarely distributed in a balanced manner. Importantly, critical or high-value data are often significantly scarce compared to those that are not. This disparity is more common in high-risk domains such as healthcare, finance, and security. From such imbalanced data, biased models are created as machine learning models lean toward majority classes while underperforming on minority classes, which are, ironically, often the most consequential.

Previous methods for handling data imbalance have challenged this issue from various angles and provided crucial insights. Methods such as HCBOU Salehi & Khedmati (2025) show how data sampling can effectively mitigate bias from imbalance, RIDE Wang et al. (2020) highlights the importance of prediction diversity in models during ensembling, and SADE Zhang et al. (2022) demonstrates how an appropriate weighting technique based on model performance can improve performance. However, common limitations such as training instability, overfitting, and trade-offs between the majority class and minority class remain.

To overcome the limitations of existing methods, we propose **BESTI**, Bayesian Class-wise Trust-Weighted Prediction Ensemble with Structured Sampling for Imbalanced multi-class classification. This framework aims to address the data imbalance issue through two core ideas: 1) Construct multiple sub-training sets that reflect diverse imbalance levels to train the models, and 2) Ensemble models based on class-wise trustworthiness by giving more influence during aggregation.

BESTI applies **Structured Sampling** to the training data. This is a framework, not a single rule, that creates sub-training datasets using threshold-based clipping: downsampling classes with more

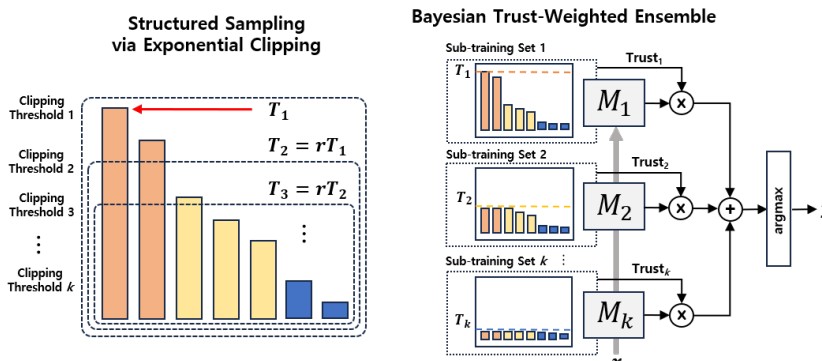

Figure 1: The overall workflow of BESTI, which includes the structured sampling via exponential decay clipping and the Bayesian class-wise trust-weighted ensemble.

instances than the threshold, while preserving others. The strategies of instantiating thresholds include Exponential Decay Clipping (EDC), Uniform Interval, Quantile Selection, and K-means-based thresholds. After independently training one model per subset, **Bayesian Class-wise Trust-Weighted Prediction Ensemble** is applied by utilizing a model's class precision as a Bayesian estimate of trust, which is the probability that a prediction is correct given the predicted class. These trust scores are then used to weight model predictions during ensembling, allowing more reliable models to have a stronger influence for specific classes.

Experiments on long-tailed datasets such as CIFAR-100-LT, ImageNet-LT, and Places-LT demonstrate BESTI's capability in handling imbalanced data. BESTI shows consistent improvements in performance throughout the dataset, along with the **flexibility** for a wide range of backbone models and diverse learning schemes. Comparison with state-of-the-art methods highlights BESTI's competitiveness despite the fact that it only **relies solely on internal data** (not requiring performance feedback-based tuning or re-weighting). Results also demonstrate that BESTI **minimizes performance trade-offs** between majority class and minority class, and many times even improves the majority classes.

## 2 RELATED WORKS

Due to its simple and scalable natures, the One-vs-All (OVA) decomposition strategy has been widely adopted in multi-class classification tasks Rifkin & Klautau (2004). However, OVA brings significant limitations in terms of severe class imbalance, since each binary task pits a single class against the remaining majority. To address this, various sampling techniques such as a combination of undersampling and oversampling have been introduced.

The Hybrid Cluster-Based Oversampling and Undersampling (HCBOU) method proposed by Salehi and Khedmati Salehi & Khedmati (2025) is a representative example. Instead of blindly or randomly oversampling and undersampling the classes, it first applies clustering methods such as k-means clustering. Based on the clusters within each class, it distinguishes valuable data to keep when undersampling or to duplicate when oversampling. This structured strategy allows HCBOU to reduce imbalance while preserving local data geometry, leading to improved classifier generalization.

While sampling methods to balance the dataset are a straightforward solution to class imbalance problems, model ensemble is a probable alternative for this issue. For example, Jabir et al. Jabir et al. (2023) proposed Ensemble Partition Sampling (EPS), which combines oversampling, minimal undersampling, and ensembling under the OVA framework. EPS creates multiple balanced binary datasets for each class and aggregates the predictions from independently trained classifiers. This leads to improved robustness and sensitivity across class frequencies.

Such ensemble-based strategies have also become prominent in long-tailed classification, a subdomain of imbalanced learning. In this context, Routing Diverse Experts (RIDE) Wang et al. (2020) and Self-supervised Aggregation of Diverse Experts (SADE) Zhang et al. (2022) are widely recog-

nized as state-of-the-art (SOTA). RIDE leverages a shared backbone with multiple classifier heads and encourages prediction diversity through a regularization term, routing predictions across heads at inference time. SADE adopts a self-adaptive ensemble mechanism that not only adjusts each model's contribution but also incorporates sample-level difficulty into the training process. By assigning greater weights to hard or underrepresented instances, ensemble steps focus on challenging regions of the data during the learning.

BESTI derives key insights from these prior studies. HCBOU exemplifies the effectiveness of sampling methods, RIDE emphasizes the importance of prediction diversity during the ensembling process, and SADE highlights the use of weighting based on difficulty or reliability. Based on these advances, BESTI integrates structured sampling with a trust-weighted ensemble framework. By generating sub-models with varying imbalance levels and combining their predictions through class-specific trust scores, BESTI aims to adopt strengths from various prior approaches for the imbalance problem in multi-class classification.

## 3 METHODOLOGY

### 3.1 OVERVIEW OF BESTI

We propose a **B**ayesian trust-weighted **E**nsemble with **ST**ructured sampling for **I**mbalanced multi-class classification, named BESTI, which brings the diversity of training data samples in models and utilizes appropriate weighting mechanisms to handle data imbalance. The BESTI framework consists of two core processes: 1) Structured sampling through threshold-based clipping and 2) Bayesian class-wise trust-weighted prediction ensemble as shown in Figure 1. The overall workflow of BESTI illustrates how an imbalanced dataset undergoes structured sampling and the Bayesian class-wise trust-weighted prediction ensemble process.

*Structured Sampling's (SS)* purpose lies in producing a sequence of sub-training sets where each set reflects a different level of class imbalance while preserving all classes in every set. This sequence of sub-training sets is constructed using threshold-based clipping. As the clipping threshold lowers in a structured manner, minority classes become more represented in sub-training sets. Compared to traditional single-set downsampling that leads to significant data loss, this leads to improved inclusivity and diversity among models used for ensembling. This approach not only preserves the structural characteristics of the original data, but also ensures fair exposure for each class within the training set group.

*Bayesian Class-wise Trust-Weighted Ensemble* aggregates multiple models that are independently trained from each sub-training set derived from SS. In this framework, a class-wise trust score is used to weight the prediction from each trained model. The trust scores are derived by Bayes' theorem from the model's confusion matrix, which simplifies to the precision metric for each class. Consequently, the ensemble prediction for each class is computed as a weighted sum of the outputs from individual models, where the weights correspond to these class-specific trust scores. If model's trust score is higher compared to other models in certain classes, its prediction for those particular classes contributes more heavily to the ensemble. When models are trained on subsets with diverse class distributions, some models naturally specialize in certain classes. These specialized models then exert greater influence on the ensemble, ultimately improving the overall accuracy.

The modularity of BESTI–decoupling structured sampling, model training, and trust-weighted aggregation–makes it efficiently parallelizable for large-scale imbalanced datasets.

### 3.2 STRUCTURED SAMPLING VIA THRESHOLD-BASED CLIPPING

BESTI is composed of multiple stages, where each stage, index $i$ represents the number of models used for ensembling. As the stage progresses, BESTI needs to generate a new sub-training set for a new model. In this process, BESTI incorporates as much diversity in imbalance level as possible through SS, since models with different perspectives are crucial for the ensembling process. The sub-training sets are generated through clipping the original training set: downsampling classes with larger instance counts than the clipping threshold while preserving classes that have equal or fewer instance counts.

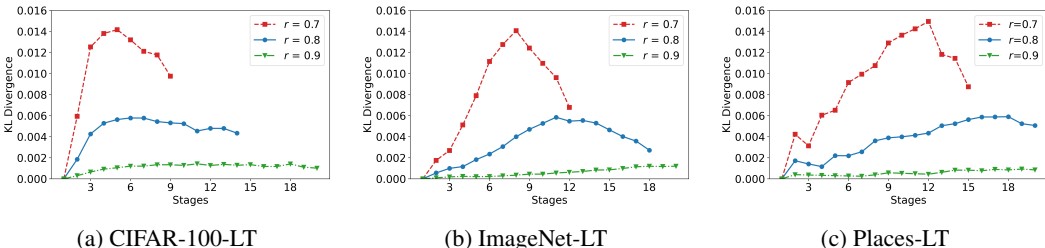

(a) CIFAR-100-LT       (b) ImageNet-LT       (c) Places-LT

Figure 2: Plots of KL divergence between clipped data sample distributions of consecutive stages for the CIFAR-100-LT, ImageNet-LT, and Places-LT datasets.

We consider four strategies for instantiating thresholds for SS: (i) **Exponential Decay Clipping (EDC)** — ceilings shrink exponentially across stages; (ii) **Uniform Interval** — thresholds equally partition the range between the minimum and maximum number of samples; (iii) **Quantile Selection** — each subset contains approximately the same number of classes; (iv) **K-means–based thresholds** — class counts are clustered, and the maximum count in each cluster defines a ceiling. Among these, we detail EDC below as a concrete example; the framework itself is agnostic to the specific schedule.

Let $T_i$ and $S_i$ denote the clipping threshold and the sub-training set for the $i$-th stage, respectively. Then, under the exponential decaying, a formalization of obtaining a new sub-training set at Stage $i$ is as follows:

$$T_i = \left\lfloor C_{\max} \times r^{i-1} \right\rfloor, \tag{1}$$

where $C_{max}$ is the largest number of samples for the original training dataset.

$$\mathcal{S}_i = \bigcup_{c \in \mathcal{C}} \text{Sample}(X_c, \min\{T_k, |X_c|\}), \tag{2}$$

where $X_c = \{x \in \mathcal{D}_{\text{train}} \mid \text{label}(x) = c\}$, and $\text{Sample}(X, n)$ represents a subset of $n$ elements randomly selected from $X$ without replacement. The progressive generation of sub-training sets with lower clipping thresholds allows later stages to train more models on increasingly balanced subsets.

Although increasing the number of stages can improve performance, it also incurs computational overhead. Therefore, finding an appropriate number of models for ensembling is a critical issue in practice. We propose using Kullback–Leibler (KL) divergence to find an appropriate number of models, i.e., the final stage index. While the KL divergence is commonly used for measuring the difference between two probability distributions, in BESTI, it is utilized for calculating the difference of class distributions between training sets. This process can be formalized as:

$$D_{KL}(P \parallel Q) = \sum_c P(c) \log \frac{P(c)}{Q(c)}, \tag{3}$$

where $P(c)$ represents the class distribution of the previous sub-training set, and $Q(c)$ represents that of the current one.

Figure 2 shows how the KL divergence changes as the stage progresses for popular long-tailed datasets, including CIFAR-100-LT, ImageNet-LT, and Places-LT. It is observed that the KL divergence rapidly increases with respect to the number of clippings and then decreases after. This implies that beyond certain stage, the class distribution does not change significantly. If the diversity of the class distribution becomes smaller, training a model on such sub-training sets would yield fewer advantages. Considering this relationship, the KL divergence is used to assess the computational cost and the advantages of having more models to define $k$, the final number of stages.

By its design, each sub-training set constructed via the threshold-based clipping includes all classes (i.e., no class is ever excluded from any training set). While early stages (e.g., Stages 1 and 2) preserve the original imbalance structure, later stages yield more balanced subsets, resulting in progressively diverse training sets that maintain full class coverage. Through training independent models from sub-training sets produced at each stage, BESTI captures varying perspectives on the data distribution, which are later combined through the Bayesian class-wise trust-weighted ensemble to form a more balanced and robust final predictor.

### 3.3 BAYESIAN INTERPRETATION OF CLASS-WISE TRUST

Let model $M_i$ predict class $c$. The probability that the true label is indeed $c$, given that $M_i$ predicted $c$, is expressed as the Bayesian posterior:

$$P_i(c) = \text{Prob}(\text{True} = c \mid \text{Predict} = c, M_i) \tag{4}$$

This posterior naturally reflects how much we can *trust* the model $M_i$'s prediction on class $c$.

**Proposition 1.** For a given model, the class-wise precision in the confusion matrix is equivalent to the Bayesian posterior probability that the model's prediction for class $c$ is correct. Therefore, precision can be directly interpreted as a class-wise trust score in Bayesian terms.

*Proof.* See Appendix C for the full proof.

**Remark 1.** Precision is not just a metric of classification performance—it is a theoretical estimate of how *trustworthy* a model is when it predicts a certain class. Especially in imbalanced classification settings, where some models specialize in certain regions of the label space, this interpretation becomes practically useful.

**Remark 2.** This proposition supports the trust-weighted ensemble strategy used in the BESTI method. By weighting each model's prediction by its class-wise trust (i.e., precision), an ensemble that naturally emphasizes more reliable models is created, and this leads to a more robust and fair prediction across all classes.

#### 3.3.1 PREDICTION AGGREGATION AND TRUST ESTIMATION

BESTI combines the outputs of multiple models through a trust-weighted ensemble strategy. For each test input $x$, each model $M_i$ produces a logit vector $z_{i,c}(x)$. These are aggregated into a final ensemble distribution by computing a class-wise weighted sum:

$$\hat{P}(y = c \mid x) = \text{softmax}\left(\sum_{i=1}^{k} w_i(c) \cdot z_{i,c}(x)\right), \tag{5}$$

where $w_i(c) = \frac{\exp(P_i(c))}{\sum_{j=1}^{k} \exp(P_j(c))}$, which serves as a trust score representing how reliable model $M_i$ is when predicting class $c$. Note that for each class c, the logits from models with higher class-wise precision are assigned greater weights. These trust scores are readily computed from confusion matrices. To avoid data leakage, we do not use the test set to compute these confusion matrices. Each model $M_i$ is evaluated on the original training dataset $\mathcal{D}_{\text{train}}$. Even though $M_i$ was trained on a structured subset $\mathcal{S}_i$, the confusion matrix for each model is constructed from largely unseen data, mitigating overfitting while maximizing the use of available samples.

After computing the ensemble distribution $\hat{P}(y = c \mid x)$, we select the final predicted label via the standard maximum a posteriori (MAP) rule, i.e., $\hat{y} = \arg\max_c \hat{P}(y = c \mid x)$. This aligns with common practice in probabilistic classifiers and ensures comparability with baseline models. The use of $\arg\max$ on trust-weighted probabilities allows BESTI to produce interpretable and decisive predictions from an otherwise soft ensemble distribution. This trust-weighted aggregation, followed by $\arg\max$ selection, completes the BESTI inference process. The detailed pseudocode is provided in Algorithm 1.

## 4 EXPERIMENTS

### 4.1 DATASET & SETUPS

We evaluate BESTI on three widely used long-tailed benchmarks: CIFAR-100 Long-Tailed Cao et al. (2019a), ImageNet-LT Liu et al. (2019), and Places-LT Liu et al. (2019). While BESTI is designed as a general-purpose method for handling data imbalance across domains, long-tailed image classification benchmarks were used for evaluation since they present particularly well-defined and challenging testbeds for multi-class imbalance.

---

**Algorithm 1** BESTI: Bayesian Trust-Weighted Ensemble with KL-based Progressive Clipping

---

**Require:** Long-tailed training dataset $\mathcal{D}_{\text{train}}$, test set $\mathcal{D}_{\text{test}}$, decay rate $r$, maximum number of stages $I$, model architecture $\mathcal{M}$
**Ensure:** Final prediction $\hat{y}$ for each $x \in \mathcal{D}_{\text{test}}$
 1: Compute instance count $n_c$ for each class $c$ in $\mathcal{D}_{\text{train}}$
 2: Initialize ceiling $T_1 \leftarrow C_{\max} = \max_c n_c$
 3: Initialize stage index $i \leftarrow 1$
 4: **for** $i = 1$ to $I$ **do**
 5:     Update threshold: $T_i \leftarrow \lfloor C_{\max} \times r^{i-1} \rfloor$
 6:     Construct sub-training set $\mathcal{S}_i$ by (2)
 7:     Train model $M_i = \text{Train}(\mathcal{M}, \mathcal{S}_i)$
 8:     Evaluate $M_i$ on $\mathcal{D}_{\text{train}}$ to compute confusion matrix $\mathbf{CM}_i$
 9:     Derive class-wise precision scores: $\text{Precision}_{i,c}$
10: **end for**
11: **for** each $x \in \mathcal{D}_{\text{test}}$ **do**
12:     **for** $i = 1$ to $I$ **do**
13:         Compute prediction probabilities $P_i(y = c|x)$ using $M_i$
14:         Obtain class-wise trust of $\text{Precision}_{i,c}$ from $\mathbf{CM}_i$
15:     **end for**
16:     Aggregate predictions using the weighted sum in (5)
17:     Final prediction by $\hat{y} = \arg\max_c \hat{P}(y = c \mid x)$
18: **end for**

---

- CIFAR-100 LT consists of 100 classes with $32\times32$ color images. The class distribution follows an exponential decay pattern governed by an imbalance ratio (IR), set to 100 in our experiments.

- ImageNet-LT is a long-tailed version of the standard ImageNet-2012 dataset, containing 1,000 classes and 186,689 training images. The class instance count ranges from 5 to 1,280, resulting in IR of 256.

- Places-LT is a long-tailed dataset from Places-2 365 dataset for scene classification. It contains 365 classes and 62,500 images. Unlike CIFAR and ImageNet, which focus on object classification, Places-LT is more focused on scene understanding under class imbalance with IR of 996.

All experiments were conducted on Ubuntu 24.04 LTS with an Intel Core i9-13900 CPU, 64GB RAM, and NVIDIA RTX 4090 GPUs (24GB VRAM each), using CUDA 12.2 and PyTorch 2.7.1.

### 4.2 BASELINES AND BACKBONES

For CIFAR-100-LT, ResNet-32 He et al. (2016), WideResNet-28 Zagoruyko & Komodakis (2016), and Vision Transformer (ViT) Dosovitskiy et al. (2021) were used to test BESTI. For ImageNet-LT and Places-LT, on the other hand, ResNeXt-50 and ResNet-152 pretrained on ImageNet were used accordingly. While a diverse range of baseline architectures, optimizers, and learning strategies were tested in all three datasets during the process, results in this paper come from models that follow configurations adopted in previous work Zhang et al. (2022) for a comparison.

Importantly, BESTI at its core is an ensembling strategy; it can flexibly integrate with more advanced weighting schemes, such as LDAM, DRW Cao et al. (2019b), and class-based loss Cui et al. (2019). However, for a fair comparison, only the basic reweighting mechanisms, such as Balanced Softmax (BSM), were used in the experiment. Note that all experiments were repeated five times with different seeds and averaged for reporting. Full details of training schedules, data augmentations, optimizers, and hyperparameter settings for each model are provided in Appendix D.

### 4.3 EVALUATION ON CIFAR-100-LT

Table 1 summarizes the performance of BESTI across three different backbones on CIFAR-100-LT with an imbalance ratio (IR) of 100. Across all models, BESTI consistently improves accuracy

Table 1: BESTI accuracy performance on CIFAR-100 LT.

(a) Ensemble Models (Exponential Decaying $r$=0.9)

| Backbone | Method | Many | Med. | Few | All |
|---|---|---|---|---|---|
| | Base (CE) | 68.6 | 39.6 | 10.7 | 41.1 |
| | BESTI ($k$=3) | 67.0 | 48.7 | 33.4 | 50.5 |
| ResNet32 | BESTI ($k$=6) | 68.5 | 50.2 | 34.5 | 51.9 |
| | BESTI ($k$=9) | 68.6 | 50.7 | 34.9 | 52.2 |
| | BESTI ($k$=12) | 68.3 | 51.4 | 35.1 | 52.4 |
| | Base (CE) | 75.3 | 44.6 | 11.9 | 45.5 |
| | BESTI ($k$=3) | 76.5 | 53.7 | 26.2 | 53.5 |
| WRN28 | BESTI ($k$=6) | 77.3 | 54.8 | 27.3 | 54.4 |
| | BESTI ($k$=9) | 77.3 | 55.3 | 28.2 | 54.9 |
| | BESTI ($k$=12) | 77.2 | 55.6 | 28.6 | 55.0 |
| | Base (CE) | 93.8 | 81.3 | 61.9 | 79.8 |
| | BESTI ($k$=3) | 94.1 | 86.8 | 78.8 | 87.0 |
| ViT | BESTI ($k$=6) | 94.5 | 87.4 | 79.5 | 87.5 |
| | BESTI ($k$=9) | 94.6 | 87.7 | 79.9 | 87.8 |
| | BESTI ($k$=12) | 94.5 | 88.0 | 80.2 | 87.9 |

(b) Clipping Methods (ResNet32, $k$=9)

| Clipping Method | Many | Med. | Few | All |
|---|---|---|---|---|
| Exponential Decay ($r$=0.9) | 68.6 | 50.7 | 34.9 | 52.2 |
| Uniform Interval | 67.2 | 51.4 | 34.5 | 51.9 |
| Quantile Selection | 59.3 | 48.9 | 31.7 | 47.4 |
| K-means Clustering | 65.6 | 51.7 | 34.4 | 51.4 |

(c) Clipping Methods (ResNet32, $k$=12)

| Clipping Method | Many | Med. | Few | All |
|---|---|---|---|---|
| Exponential Decay ($r$=0.9) | 68.3 | 51.4 | 35.1 | 52.4 |
| Uniform Interval | 67.6 | 52.2 | 34.7 | 52.3 |
| Quantile Selection | 60.1 | 49.5 | 32.1 | 48.0 |
| K-means Clustering | 66.1 | 51.8 | 33.8 | 51.4 |

Table 2: Top-1 accuracy (%) comparison on ImageNet-LT and Places-LT.

| Method | ImageNet-LT ($r$=0.9) | | | | Places-LT ($r$=0.7) | | | |
|---|---|---|---|---|---|---|---|---|
| | Many | Med. | Few | All | Many | Med. | Few | All |
| BESTI ($k$=3) | 67.1 | 53.3 | 37.7 | 56.5 $_{\pm 0.43}$ | 43.3 | 40.9 | 34.9 | 40.6 $_{\pm 0.23}$ |
| BESTI ($k$=6) | 67.9 | 54.5 | 38.7 | 57.6 $_{\pm 0.20}$ | 43.4 | 41.2 | 35.3 | 40.9 $_{\pm 0.13}$ |
| BESTI ($k$=9) | 68.2 | 54.9 | 39.5 | 58.0 $_{\pm 0.18}$ | 43.6 | 41.3 | 35.5 | 41.0 $_{\pm 0.13}$ |
| BESTI ($k$=12) | 68.4 | 55.0 | 39.4 | 58.1 $_{\pm 0.13}$ | 43.4 | 41.8 | 35.8 | 41.2 $_{\pm 0.09}$ |
| BESTI ($k$=15) | 68.5 | 55.0 | 39.6 | 58.2 $_{\pm 0.10}$ | 43.1 | 42.3 | 36.2 | 41.4 $_{\pm 0.09}$ |

for all many-, medium-, and few-shot groups and achieves the best performance at the largest $k = 12$. The most significant gains were observed in the medium- and few-shot class groups, which are the most challenging regions in long-tailed classification. For instance, BESTI with ResNet32 at $k = 12$ achieves a $+11.3\%$ improvement in macro accuracy with a $+11.8\%$ gain in medium classes and $+24.4\%$ in few-shot compared to the baseline model trained with a cross-entropy (CE) loss. As shown in Table 1a, these gains are not only consistent across backbone architectures but also robust to different degrees of baseline performance (e.g., from ResNet32 to ViT). Moreover, the fact that BESTI improves strong models such as ViT (e.g., $+8.1\%$ in macro accuracy) indicates that its benefits are complementary to architectural advancements and loss reweighting schemes (e.g., Balanced Softmax).

Tables 1b and 1c show the accuracy performance for different clipping methods for $k = 9$ and 12, respectively. As CIFAR100-LT shows a smoothly decaying distribution of class samples, the exponential decay and uniform interval methods achieve higher accuracy. Due to its long-tailed characteristics, quantile selection produces sub-datasets with extremely low thresholds, resulting in lower performance. If the distribution of class samples is highly irregular, the k-means clustering method is expected to perform better. These results validate that BESTI is universally applicable across architectures, scalable across imbalance severity levels, and particularly effective in amplifying performance in underrepresented class regions.

## 4.4 EVALUATION ON IMAGENET-LT AND PLACES-LT

Table 2 shows how macro, many-, medium-, and few-shot accuracy change over stages for ImageNet-LT and Places-LT. Considering the imbalance ratio of each dataset, ImageNet-LT used 0.9 as the decay rate and 0.7 for Places-LT. For both ImageNet-LT and Places-LT, the macro accuracy gradually increases as the stage proceeds, proving BESTI's ability to generalize over different domains. For instance, ImageNet-LT showed $+1.7\%$ increase in macro accuracy between $k = 3$ and $k = 15$. Similar to the behavior in CIFAR-100 LT, the increase in $k$ resulted in improve-

Table 3: Top-1 accuracy (%) comparison across CIFAR100-LT, ImageNet-LT, and Places-LT across many-, medium-, and few-shot categories. BESTI is compared with state-of-the-art baselines.

| Method | CIFAR100-LT | | | | ImageNet-LT | | | | Places-LT | | | |
|---|---|---|---|---|---|---|---|---|---|---|---|---|
| | Many | Med. | Few | All | Many | Med. | Few | All | Many | Med. | Few | All |
| Softmax (CE) | 68.6 | 39.6 | 10.7 | 41.4 | 68.1 | 41.5 | 14.0 | 48.0 | 46.2 | 27.5 | 12.7 | 31.4 |
| Causal Tang et al. (2020) | 64.1 | 46.8 | 19.9 | 45.0 | 64.1 | 45.8 | 27.2 | 50.3 | 23.8 | 35.7 | 39.8 | 32.2 |
| Balanced Softmax Ren et al. (2020) | 59.5 | 45.4 | 30.7 | 46.1 | 64.1 | 48.2 | 33.4 | 52.3 | 42.6 | 38.2 | 32.7 | 39.4 |
| MiSLAS Zhong et al. (2021) | 60.4 | 49.6 | 26.6 | 46.8 | 62.0 | 49.1 | 32.8 | 51.4 | 41.5 | 39.5 | 32.7 | 37.6 |
| LADE Hong et al. (2021) | 58.7 | 45.8 | 29.8 | 45.6 | 64.4 | 47.7 | 34.3 | 52.3 | 42.4 | 32.3 | 39.2 | 39.2 |
| RIDE Wang et al. (2020) | 67.4 | 49.5 | 23.7 | 48.0 | 68.0 | 52.9 | 35.1 | 56.3 | 43.1 | 41.0 | 33.0 | 40.3 |
| SADE Zhang et al. (2022) | 65.4 | 49.3 | 29.3 | 49.8 | 66.5 | 57.0 | 43.5 | 58.8 | 40.4 | 43.2 | 36.8 | 40.9 |
| **BESTI (ours)** | 68.3 | 51.4 | 35.1 | 52.4 | 68.5 | 55.0 | 39.6 | 58.2 | 43.1 | 42.3 | 36.2 | 41.4 |

ments especially in medium- and few-shot groups for both datasets. Between $k = 3$ and $k = 15$, few-shot accuracy increased by 1.3 percentage points on Places-LT and by 1.9 percentage points on ImageNet-LT. Notably, on ImageNet-LT, the increase in $k$ did not harm the performance of the many-shot group at all, but further improved its accuracy. For Places-LT, which is considered the most difficult dataset among three datasets used in the experiment, showed consistent improvements in medium-, few-shot, and macro accuracy while maintaining comparable performance in many-shot group.

## 4.5 COMPARISONS WITH SOTA

Table 3 compares BESTI with a range of state-of-the-art methods on CIFAR-100-LT, ImageNet-LT, and Places-LT, grouped by many-, medium-, and few-shot performance. BESTI shows the highly competitive macro-accuracy across all three datasets. Importantly, BESTI mitigates the crucial problem of trade-offs, which many existing approaches, such as MiSLAS, LADE, RIDE, and SADE suffer from. Our framework improved performance on medium or few-shot classes at the minimal expense of many-shot accuracy. In fact, for CIFAR-100-LT and ImageNet-LT, BESTI not only minimized but brought notable improvements to all class groups, unlike prior methods that had lower many-shot accuracy than baseline models.

For CIFAR-100-LT, BESTI outperformed previous methods in all groups. When compared with SADE, it shows $2.9\%$ increase in many-, $2.1\%$ increase in medium-, and $5.8\%$ increase in the few-shot group. According to Table 1, BESTI with only $k$=3 models outperforms the SOTA methods in terms of macro accuracy. In the case of ImageNet-LT, BESTI demonstrates dominant performance in many-shot accuracy with competitive medium-shot accuracy, leading to competitive macro accuracy. While BESTI does not show significant performance for individual shot groups in Places-LT, it achieves the highest macro accuracy over all other methods, implying that the sub-training set is formed in a balanced manner, preserving and boosting different parts of the data in an appropriate way.

Practically, continuously increasing the number of stages can improve the accuracy on few-shot classes, since more models that better represent these classes will be included in the ensemble stage. One of BESTI's greatest novelties is that it achieves such performance improvements without relying on additional validation or test datasets, whereas prior ensemble-based SOTA methods such as RIDE Wang et al. (2020) and SADE Zhang et al. (2022) relied heavily on re-weighting and performance-feedback-based tuning, which incorporate external data into the process.

## 5 ABLATION STUDIES

Table 4 reports the macro, many-, medium-, and few-shot accuracy of the loss functions and ensemble methods of key BESTI components in the ablation study. As the BESTI trains each model using the BSM loss function, it can achieve higher performance for the imbalanced dataset. We replaced the BSM loss function with the cross entropy (CE) function. In this case, the macro accuracy dropped severely to 46.2%, which is yet much higher than the baseline performance (i.e., 41.1% in Table 1) that is achieved with the single model trained with CE. In Table 3, Balanced Softmax

Table 4: Ablation study on CIFAR-100-LT dataset for ResNet32 model.

| Method ($k$=9) | Many | Med. | Few | All |
|---|---|---|---|---|
| BESTI (ours, $r$=0.9) | 68.6 | 50.7 | 34.9 | 52.2% $_{\pm 0.52\%}$ |
| w/o BSM (i.e., CE loss) | 74.9 | 45.8 | 13.1 | 46.2% $_{\pm 0.35\%}$ |
| with Naive Ensemble | 69.7 | 50.9 | 32.4 | 51.9% $_{\pm 0.55\%}$ |
| w/o BSM, Class-wise Trust | 74.9 | 45.0 | 11.4 | 45.4% $_{\pm 0.37\%}$ |

obtained 46.1% macro accuracy with 30.7% few-shot accuracy. This result indicates that the class-wise trust with SS using threshold-based clipping achieves a +6.1% gain in macro accuracy and a +3.6% gain in few-shot accuracy compared to Balanced Softmax. Note that BESTI does not rely on a specific backbone or loss function, and can flexibly integrate diverse backbone models trained with different loss functions in an ensemble manner. Without Bayesian class-wise trust weighting, a 0.3% drop in macro accuracy and a 2.5% drop in few-shot accuracy are observed with the naive ensemble, which performs uniform averaging of softmax outputs. This result indicates that the class-wise trust ensemble increases the weights of the model's class outputs that yield higher precision in a class-wise manner.

## 6 LIMITATIONS

One limitation of BESTI lies in the trade-off between computational cost and performance. Increasing the number of stages or adding more models from diverse imbalance ratios can improve accuracy, but also incurs additional overhead. For example, applying a weak decay rate in EDC may generate many highly specialized models, which is not cost-efficient in practice. Therefore, depending on the sample distribution and the baseline model, hyperparameters such as $r$ and $k$ should be carefully chosen. Such limitations can lead to possible future works, such as creating automatic hyperparameter adaptation and noise-resilient estimation methods.

## 7 CONCLUSION

In this paper, we introduced BESTI, a framework for imbalanced multi-class classification. BESTI combines structured sampling with a Bayesian class-wise trust-weighted ensemble. Through the structured sampling, BESTI creates sub-training sets of various imbalance levels, allowing diversity for the models while maintaining the balance between boosting minority classes and preserving performance on majority classes. By interpreting class-wise precision as Bayesian Trust score, models independently trained from sub-training sets are aggregated in a way that more reliable predictions have a stronger influence in making the final prediction.

Experiment results from CIFAR-100-LT, ImageNet-LT, and Places-LT highlight BESTI's flexibility in different backbone models, consistent performance improvements across different shot groups, and competitiveness in overall accuracy for all three datasets. Importantly, BESTI challenges the idea of a performance trade-off between majority classes and minority classes, which has been a common yet significant shortcoming of previous methods.

In the future, BESTI can be further developed and applied in various ways. For structured sampling, methods such as KL divergence-based dynamic alteration of decay rate can be tested. Furthermore, utilizing BESTI's flexibility by ensembling heterogeneous backbone models with different learning schemes may lead to promising results. Change in domain is another option, applying BESTI to tasks with server class imbalances, such as natural language processing, tabular datasets, fraud detection, and recommendation systems.

In conclusion, BESTI, a new framework for addressing multi-class imbalance, demonstrates a combination of structured sampling and trust-weighted ensemble working together to create more balanced and robust models. Its simple, principled, and modular structure will allow BESTI to evolve with advances in classification models and learning schemes, providing a foundation for future research in imbalance learning.

## REPRODUCIBILITY STATEMENT

All implementation details are provided in Sections 4–5. We release the code as supplementary material, and the configuration used to produce all results, including dataset preprocessing, thresholds, and hyperparameters, is specified in Appendix D.

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

## ETHICS STATEMENT

This work does not involve human subjects or sensitive data. The datasets used are publicly available under permissive licenses.

Large language models (LLMs) were used only for minor language editing. All technical content was developed by the authors.

The authors declare that they have no competing interests

## A    Long-tailed Datasets

We evaluate our method on three long-tailed datasets: CIFAR-100-LT, ImageNet-LT, and Places-LT. Although all exhibit long-tailed characteristics, their class distributions differ significantly. Figure A.1 illustrates the distribution of classes in each dataset.

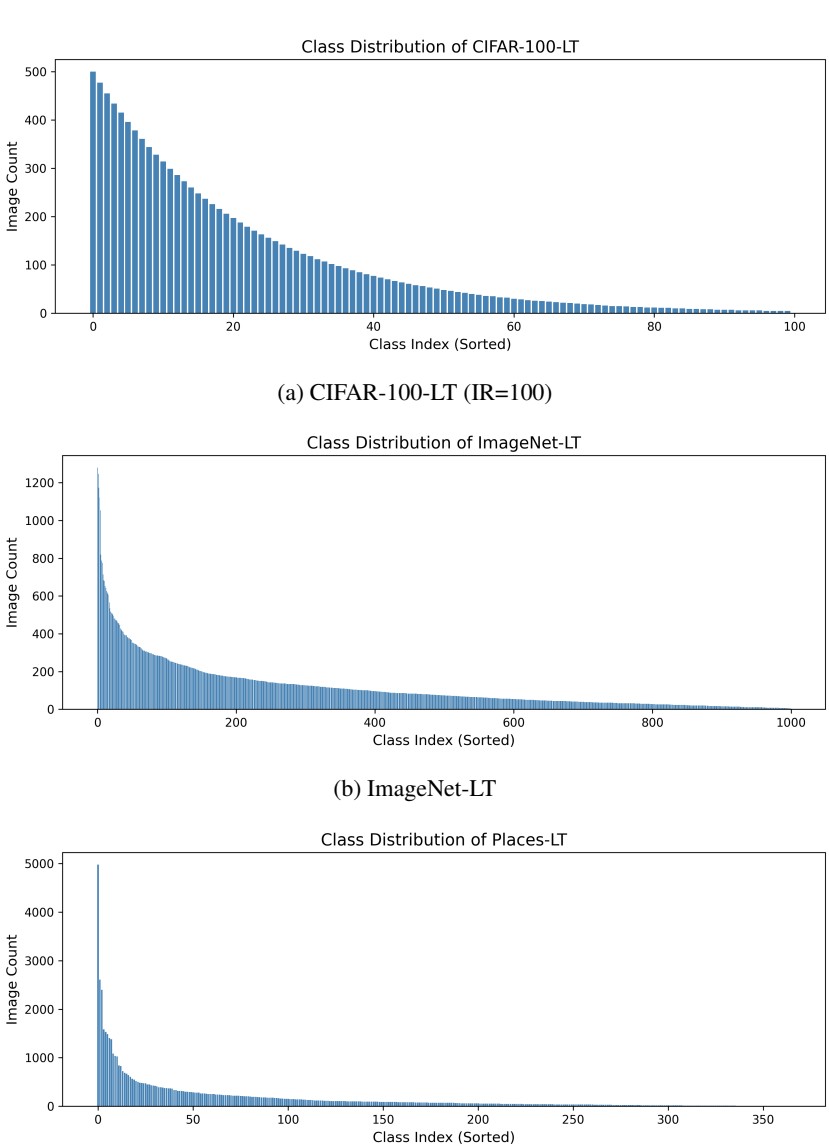

(a) CIFAR-100-LT (IR=100)

(b) ImageNet-LT

(c) Places-LT

Figure A.1: Instance count per class across CIFAR-100-LT, ImageNet-LT, and Places-LT.

As shown in the figures above, there exists a high disparity not only in terms of imbalance ratio, but also in the overall shape of the distribution across the three datasets. While CIFAR-100-LT exhibits smooth decay without significant count differences between one class and the previous class, ImageNet-LT and Places-LT show extreme concentration in the many-shot group, making the classification task more difficult.

## B   CLIPPING METHODS

Among the four clipping methods, we focused on exponential decay with a ratio $r$. The advantage is that as the number of models increases, the previously generated models can be reused without retraining all models. By training only one additional model, the ensemble size can be expanded. In contrast, for example, the uniform interval method requires training an entirely new set of models whenever the number of models $k$ changes. To facilitate evaluation and reporting, we primarily adopt exponential decay in this work. However, it should be noted that exponential decay does not necessarily yield the best performance.

As shown in Fig A.1, similar to CIFAR-100-LT, where the class distribution follows a smooth decay, exponential decay or uniform interval–based methods are generally suitable for setting clipping thresholds. In contrast, when only a few classes dominate with an extremely large number of samples, repeated decays from the largest counts fail to increase the diversity of class distributions, leading to a small KL divergence. In these situations, quantile-selection or clustering-based methods are more effective.

Table A.1 presents the performance results on the CIFAR100-LT dataset with respect to the number of ResNet32 models in the ensemble when the thresholds are generated with $k = 12$. As shown in Table 3, the SOTA algorithms achieve macro accuracy below 50% at their best configurations on the CIFAR100-LT dataset. In contrast, our proposed BESTI consistently exceeds the 50% macro accuracy threshold as early as $i = 3$, regardless of the clipping method used.

Thresholds vary considerably across clipping methods, with quantile-selection and k-means clustering yielding very small thresholds at the final stage. For example, $T_{12}$ is 8 and 14 for quantile-selection and k-means clustering, respectively, due to the long-tailed distribution of the CIFAR100-LT dataset. Notably, even with $k = 12$, the best performance is not obtained at the last stage for the two methods. This occurs because training with subsets defined by extremely small thresholds reduces overall accuracy, thereby degrading ensemble performance.

Table A.1: Performance results on CIFAR100-LT with respect to the number of ResNet32 models in the ensemble when the thresholds are generated with $k=12$.

| Clipping Method | Up to the $i$-th model | Many | Med. | Few | All |
|---|---|---|---|---|---|
| Exponential Decay ($r=0.9$) | $i=1, T_i=500$ | 62.2 | 43.3 | 27.4 | 45.1 $_{\pm 0.63}$ |
| | $i=2, T_i=450$ | 65.6 | 47.2 | 31.8 | 49.0 $_{\pm 0.70}$ |
| | $i=3, T_i=405$ | 67.0 | 48.7 | 33.4 | 50.5 $_{\pm 0.64}$ |
| | $i=4, T_i=364$ | 67.7 | 49.4 | 34.3 | 51.3 $_{\pm 0.58}$ |
| | $i=5, T_i=328$ | 68.0 | 49.8 | 34.6 | 51.6 $_{\pm 0.57}$ |
| | $i=6, T_i=295$ | 68.4 | 50.2 | 34.5 | 51.8 $_{\pm 0.55}$ |
| | $i=7, T_i=266$ | 68.6 | 50.5 | 34.8 | 52.1 $_{\pm 0.63}$ |
| | $i=8, T_i=239$ | **68.8** | 50.7 | 35.0 | 52.3 $_{\pm 0.51}$ |
| | $i=9, T_i=215$ | 68.6 | 50.7 | 34.9 | 52.2 $_{\pm 0.52}$ |
| | $i=10, T_i=194$ | 68.5 | 50.9 | **35.1** | 52.3 $_{\pm 0.55}$ |
| | $i=11, T_i=174$ | 68.6 | 51.1 | 35.0 | 52.4 $_{\pm 0.60}$ |
| | $i^*=12, T_i=\mathbf{157}$ | 68.3 | **51.4** | 35.1 | **52.4** $_{\pm 0.71}$ |
| Uniform Interval | $i=1, T_i=500$ | 62.1 | 43.3 | 26.8 | 44.9 $_{\pm 0.53}$ |
| | $i=2, T_i=459$ | 65.5 | 47.4 | 30.4 | 48.6 $_{\pm 0.88}$ |
| | $i=3, T_i=418$ | 67.0 | 48.6 | 32.3 | 50.2 $_{\pm 0.83}$ |
| | $i=4, T_i=377$ | 67.5 | 49.7 | 33.6 | 51.1 $_{\pm 0.90}$ |
| | $i=5, T_i=336$ | 68.3 | 49.9 | 34.2 | 51.6 $_{\pm 0.60}$ |
| | $i=6, T_i=295$ | **68.4** | 50.6 | 34.6 | 52.0 $_{\pm 0.65}$ |
| | $i=7, T_i=254$ | 68.4 | 50.6 | **34.8** | 52.1 $_{\pm 0.83}$ |
| | $i=8, T_i=213$ | 68.4 | 51.1 | 34.5 | 52.2 $_{\pm 0.70}$ |
| | $i=9, T_i=172$ | 68.2 | 51.1 | 34.5 | 52.1 $_{\pm 0.67}$ |
| | $i=10, T_i=131$ | 68.2 | 51.5 | 34.6 | 52.3 $_{\pm 0.59}$ |
| | $i=11, T_i=90$ | 67.9 | 51.7 | 34.7 | 52.3 $_{\pm 0.80}$ |
| | $i^*=12, T_i=\mathbf{49}$ | 67.6 | **52.2** | 34.7 | **52.3** $_{\pm 0.79}$ |
| Quantile Selection | $i=1, T_i=500$ | 62.2 | 43.6 | 27.4 | 45.2 $_{\pm 0.53}$ |
| | $i=2, T_i=344$ | 66.0 | 47.3 | 31.4 | 49.1 $_{\pm 0.44}$ |
| | $i=3, T_i=237$ | **66.9** | 48.6 | 32.4 | 50.1 $_{\pm 0.69}$ |
| | $i=4, T_i=163$ | 66.8 | 49.6 | 32.4 | 50.5 $_{\pm 0.90}$ |
| | $i^*=5, T_i=\mathbf{112}$ | 66.2 | 50.4 | 32.4 | **50.5** $_{\pm 0.85}$ |
| | $i=6, T_i=77$ | 65.3 | 50.6 | 32.4 | 50.3 $_{\pm 0.94}$ |
| | $i=7, T_i=53$ | 64.6 | **51.1** | **32.7** | 50.3 $_{\pm 0.91}$ |
| | $i=8, T_i=36$ | 63.7 | 50.8 | 32.7 | 49.9 $_{\pm 0.99}$ |
| | $i=9, T_i=25$ | 63.0 | 50.6 | 32.6 | 49.6 $_{\pm 1.07}$ |
| | $i=10, T_i=17$ | 61.9 | 50.3 | 32.1 | 48.9 $_{\pm 0.78}$ |
| | $i=11, T_i=12$ | 61.1 | 49.8 | 32.2 | 48.5 $_{\pm 0.89}$ |
| | $i=12, T_i=8$ | 60.1 | 49.5 | 32.1 | 48.0 $_{\pm 0.82}$ |
| K-means Clustering | $i=1, T_i=500$ | 62.7 | 43.8 | 27.3 | 45.4 $_{\pm 0.66}$ |
| | $i=2, T_i=434$ | 65.8 | 47.7 | 31.3 | 49.1 $_{\pm 0.65}$ |
| | $i=3, T_i=361$ | 67.2 | 49.0 | 32.2 | 50.3 $_{\pm 0.50}$ |
| | $i=4, T_i=286$ | 67.6 | 49.3 | 33.7 | 51.0 $_{\pm 0.80}$ |
| | $i=5, T_i=226$ | 68.2 | 49.7 | 33.9 | 51.4 $_{\pm 0.74}$ |
| | $i=6, T_i=179$ | **68.2** | 50.3 | 34.1 | 51.7 $_{\pm 0.62}$ |
| | $i^*=7, T_i=\mathbf{135}$ | 68.2 | 50.9 | **34.2** | **51.9** $_{\pm 0.57}$ |
| | $i=8, T_i=102$ | 68.1 | 50.9 | 33.8 | 51.8 $_{\pm 0.72}$ |
| | $i=9, T_i=70$ | 67.6 | 51.0 | 34.1 | 51.7 $_{\pm 0.59}$ |
| | $i=10, T_i=46$ | 67.0 | 51.5 | 33.7 | 51.6 $_{\pm 0.76}$ |
| | $i=11, T_i=27$ | 66.8 | 51.6 | 33.9 | 51.6 $_{\pm 0.59}$ |
| | $i=12, T_i=14$ | 66.1 | **51.8** | 33.8 | 51.4 $_{\pm 0.68}$ |

## C    PROOF OF PROPOSITION 1

By Bayes' theorem, the Bayesian posterior in (4) becomes

$$P_i(c) = \frac{P(\text{Predict} = c \mid \text{True} = c, M_i) \cdot P(\text{True} = c)}{P(\text{Predict} = c \mid M_i)},$$

where each term can be estimated from the confusion matrix of $M_i$:

$$P(\text{Predict} = c \mid \text{True} = c, M_i) = \frac{TP_{i,c}}{TP_{i,c} + FN_{i,c}} \quad \text{(Recall)}$$

$$P(\text{True} = c) = \frac{TP_{i,c} + FN_{i,c}}{N} \quad \text{(Class Prior)}$$

$$P(\text{Predict} = c \mid M_i) = \frac{TP_{i,c} + FP_{i,c}}{N}.$$

Substituting these into the Bayes expansion, we are given the following:

$$P_i(c) = \frac{TP_{i,c}}{TP_{i,c} + FP_{i,c}},$$

which corresponds to $\text{Precision}_{i,c}$. This completes the proof.

# D FULL HYPERPARAMETERS FOR EXPERIMENTS

Considering various factors such as data structure, image characteristics, and comparability with SOTA methods, the backbone and hyperparameters for each dataset were carefully selected. Table A.2 provides detailed information on the backbone models and corresponding hyperparameters used in our experiments.

Table A.2: Model configurations used across datasets.

| Dataset | Model | Opt | LR | WD | Loss | Sched | Epochs | Batch Size |
|---------|-------|-----|-----|-----|------|-------|--------|------------|
| CIFAR-100 LT | ResNet32 | SGD | 0.1 | 5e-4 | CE / BSM | MultiStep | 200 | 128 |
| | WRN-28 | SGD | 0.1 | 5e-4 | CE / BSM | MultiStep | 200 | 128 |
| | ViT-B/16 | AdamW | 5e-5 | 0.05 | CE / BSM | Cosine | 200 | 128 |
| ImageNet-LT | ResNet50 | SGD | 0.025 | 5e-4 | CE / BSM | Cosine | 180 | 64 |
| Places-LT | ResNet152 | SGD | 0.001 / 0.01 | 4e-4 | CE / BSM | LinearLR | 30 | 128 |

Detailed Python code is provided in Listing A.1.

```python
if "vit" in args.model.lower():
    optimizer = torch.optim.AdamW(model.parameters(), lr=5e-5,
        weight_decay=0.05)
    scheduler = WarmupCosineAnnealingLR(optimizer, total_epochs=args.
        epochs, warmup_epochs=5, eta_min=0.0)
elif "resnext50" in args.model.lower():
    optimizer = torch.optim.SGD(model.parameters(), lr=0.025, momentum
        =0.9, weight_decay=5e-4, nesterov=True)
    scheduler = WarmupCosineAnnealingLR(optimizer, total_epochs=args.
        epochs, warmup_epochs=5, eta_min=0.0)
elif 'resnet152' in args.model.lower():
    optimizer = torch.optim.SGD([
        {'params': [p for name, p in model.named_parameters() if not name
            .startswith("fc.")], 'lr': 0.001},
        {'params': list(model.fc.parameters()), 'lr': 0.01}
    ], momentum=0.9, weight_decay=4e-4, nesterov=True)

    warmup_scheduler = LinearLR(optimizer, start_factor=0.01, end_factor
        =1.0, total_iters=5)
    main_scheduler = LinearLR(optimizer, start_factor=1.0, end_factor
        =0.0, total_iters=args.epochs-5)
    scheduler = ChainedScheduler([warmup_scheduler, main_scheduler])
else:
    optimizer = torch.optim.SGD(model.parameters(), lr=0.1, momentum=0.9,
         weight_decay=5e-4, nesterov=True)
    scheduler = WarmupMultiStepLR(optimizer, milestones=[160, 180], gamma
        =0.1, warmup_epochs=5)
```

Listing A.1: Optimizer and scheduler settings for different backbones.

