# OpenReview forum: "BESTI: BAYESIAN CLASS-WISE TRUST-WEIGHTED ENSEMBLE WITH STRUCTURED SAMPLING FOR IMBALANCED MULTI-CLASS CLASSIFICATION"
_ICLR.cc/2026/Conference — Submitted to ICLR 2026_

### Official Review · Reviewer_qFFk · 2025-10-24

**Soundness:** 2
**Presentation:** 3
**Contribution:** 2
**Rating:** 2
**Confidence:** 4

**Summary:**

This paper proposes BESTI — a Bayesian Class-wise Trust-Weighted Ensemble framework with structured sampling for imbalanced multi-class classification. The method generates multiple sub-training sets with different imbalance levels via threshold-based clipping, trains a separate model for each subset, and aggregates predictions based on class-wise precision interpreted as a Bayesian trust score. Experiments on CIFAR100-LT, ImageNet-LT, and Places-LT claim the improvement of the proposed framework BESTI over a basic CE baseline and competitive performance against existing methods.

**Strengths:**

(1) This work introduces a structured sampling scheme that generates multiple sub-training datasets with varying imbalance levels, offering a novel perspective compared to conventional single-set resampling approaches.

(2) It provides a Bayesian interpretation of class-wise precision as trust, giving a theoretically grounded weighting mechanism in ensemble learning.

(3) The methodology is clearly stated, with intuitive motivation and mathematical formalization.

**Weaknesses:**

(1) The idea of generating multiple resampled subsets and ensemble models has been widely explored. The key novelty (Bayesian weighting using precision) is conceptually simple and may be seen as a straightforward reinterpretation rather than a fundamentally new algorithmic contribution.

(2) Precision is affected by class priors and sample imbalance, which can cause unstable weights for very rare classes. The paper does not analyze risk amplification when precision is unreliable or noisy.

(3) Only image classification benchmarks are evaluated, despite claims that BESTI benefits high-risk domains (e.g., healthcare, finance).

(4) Reported tables appear to be single-run results, without standard deviation, confidence intervals, or statistical tests. This makes the reliability of model improvements unclear, especially because sampling randomness can heavily influence performance in imbalanced settings.

(5) Only accuracy-based metrics are reported. For imbalanced classification, macro-F1, balanced accuracy, and AUC are essential, yet largely missing. Include class-balanced metrics to more appropriately measure impact on few-shot classes.

(6)  Experimental evidence is insufficient to fully support the claims. For example, in Table 3, the proposed method does not demonstrate a clear advantage compared to the baseline methods.

**Questions:**

The following are my detailed questions and concerns:

(1) It is unclear why structured thresholds produce better class representation diversity than a single optimized sampling strategy. No ablation is shown to justify the design.

(2) The claim that models trained with different imbalance levels specialize in specific classes requires empirical proof (e.g., per-class performance visualization).

(3) The authors claim parallelizability, but do not show actual parallel performance gains or resource requirements.

(4) High precision does not guarantee good minority recall. Why is precision chosen over recall in imbalanced settings? The motivation of using precision is not clear.

(5) Improvements are reported as single values without statistical testing (e.g., variance / confidence intervals). Are results consistent across multiple random seeds or different random dataset split?

(6) Since sampling is random, how is stability ensured? Are multiple runs averaged or is there high variance in created subsets and trained models? For example in table 1, why not report the mean and std results?

(7) Dataset subsets progressively discard majority class data. How does this affect learning of decision boundaries? Does performance degrade with too many stages?

(8) In table 3, the performance of the proposed BESTI does not demonstrate an advantage over other baselines, especially in Many, Med, Few group on ImageNet-LT and Places-LT. How to claim that BESTI performs better than others? And again, why not run multiple times and report the average performance?

---

### Official Review · Reviewer_DQbA · 2025-10-27

**Soundness:** 2
**Presentation:** 2
**Contribution:** 1
**Rating:** 2
**Confidence:** 4

**Summary:**

This paper introduces a framework called BESTI for handling imbalanced multi-class classification based on three components: i) generating multiple training subsets with structured sampling to create diverse models, ii) training independent models on these subsets, and iii) aggregating their predictions using a Bayesian trust-weighting scheme where each model's class-wise precision determines its influence. Experiments on long-tailed image benchmarks (CIFAR-LT, ImageNet-LT) show that BESTI effectively improves overall accuracy, particularly for minority classes, and achieves competitive performance with state-of-the-art methods.

**Strengths:**

-  Experiments conducted on 3 challenging datasets show the method consistently demonstrates improved performance, particularly for the crucial medium- and few-shot classes. Although this is something that has been studied and demonstrated for years in ensemble learning for imbalanced problems.
- The decoupling of "Structured Sampling" (generation) and "Trust-Weighted Ensemble" (prediction) is a great thing as it allows users to use each module independent of the other, and is also aligned with the ensemble literature which divides a whole ensemble into generation, selection and aggregation
- source code is available as supplementary material which is good for reproducibility

**Weaknesses:**

**Major weaknesses:**
- Lines 314–316 state that “BESTI … can flexibly integrate with more advanced weighting schemes, such as LDAM, DRW [Cao et al., 2019b], and class-based loss [Cui et al., 2019].” This flexibility contradicts the paper’s central claim of a Bayesian class-wise trust-weighted combination rule. If external weighting schemes replace or modify the proposed trust weights, the “Bayesian Trust” principle is no longer preserved.

- The claim that BESTI introduces a “Bayesian Class-wise Trust-Weighted Ensemble” and its novelty is quite overstated. Computing per-class precision and using it as a weighting factor is conceptually equivalent to earlier local accuracy estimation and trainable combiner approaches [1][2]. Those works already treat class-specific performance as a competence estimate for combination weights. The paper should demonstrate a theoretical advancement beyond these models or tone down the “Bayesian” framing, which currently lacks derivation or probabilistic justification.

- The proposed Structured Sampling (SS) strategy resembles existing resampling-based ensembles such as Random Balance [3][4] and related diversity-enhancing methods [5]. The paper should clarify how SS fundamentally differs from these techniques beyond using a deterministic threshold schedule and include empirical or theoretical justification for calling it new. Also, why not consider oversampling the minority classes together as an approach to handle this problem?

- The manuscript includes strong claims about the method’s advantages: “incorporates as much diversity in imbalance level as possible” and produces “interpretable and decisive predictions” , without supporting analysis. The first statement is inaccurate: the Structured Sampling procedure only reduces majority classes progressively, never generating subsets where minority classes become dominant. Prior ensemble works on imbalance (e.g., Random Balance [3][4][5]) explicitly vary class priors in both directions and empirically show that such bidirectional sampling increases robustness. Thus, BESTI does not maximize diversity in class distributions, but rather explores a restricted region of the sampling space. The second statement on “interpretable and decisive predictions” is generic to any argmax-based soft ensemble [6]. These claims should be rephrased or substantiated with empirical or theoretical justification.

- Section 2 begins with One-vs-All (OVA) classification, which disrupts the flow of the imbalance discussion. OVA can induce imbalance, but BESTI’s setting naturally long-tailed datasets is independent of that decomposition. The section should open with long-tailed [8] and ensemble literature instead, mentioning OVA only briefly as background.

- Diversity is measured only via KL divergence between training-set class distributions, not via prediction-level behavior. In ensemble learning, what matters is complementarity of predictions [7]; training differences are only a proxy. The paper should quantify prediction-level diversity (e.g., pairwise disagreement, Q-statistic, co-error rate, double fault) to verify that experts truly specialize rather than replicate one another.

- The text claims the approach is “parallelizable and scalable,” yet Algorithm 1 and the staged sampling description imply sequential dependency (stage 1, 2, …) implying later models depend on progressively clipped subsets. The authors should clarify how models are generated and whether they can indeed be trained in parallel as well as reflect that in the writing.

- In computing per-class precision, each model M_iis evaluated on D_train, which partially overlaps with its training subset S_i. This compromises the claim of using “largely unseen data.” The paper should quantify the overlap and discuss its potential impact on the trust estimation.
	Different backbones are used for different datasets without justification. For xcomplete comparison, each backbone should be tested across all datasets or clear justification for not using them (e.g., computational limits, incompatibility) should be provided. Similarly, it is not specified which weighting variants, optimizers etc were tested on which backbones, leaving the methodology ambiguous.

-Although the limitations section mentions efficiency, the paper provides no quantitative analysis of computational cost as the ensemble grows . Since inference cost scales linearly with ensemble size, a cost–performance curve or per-model accuracy gain would strengthen the empirical evaluation.

- The ablation studies do not fully isolate the contributions of BESTI’s three core components: Structured Sampling, independent model training, and Trust-Weighted Aggregation. For instance, the paper reports variants such as “w/o BSM” or “Naive Ensemble,” but never evaluates the effect of removing Structured Sampling itself or replacing it with a known baseline (e.g., Random Balance). As a result, it remains unclear how much of the improvement comes from each module. Since the paper emphasizes BESTI’s modularity and “efficient parallelizability” , a comprehensive ablation should be done.

**Minor presentation issues:**
boldfacing best scores on tables (or using colors etc)  would help the reader finding trends in the results. Also, the authors do not mention the code is available as supplementary material in the open-review website. Some readers may miss that.

**References**

[1] Woods, K., Kegelmeyer, W. P., & Bowyer, K. (1997). Combination of multiple classifiers using local accuracy estimates. IEEE TPAMI 19(4), 405–410.
[2] Kuncheva, L. I. (2014). Combining Pattern Classifiers: Methods and Algorithms. Wiley.
[3] Díez-Pastor, J. F., et al. (2015). Random balance: ensembles of variable priors classifiers for imbalanced data. Knowledge-Based Systems 85, 96–111.
[4] Rodríguez, J. J., et al. (2020). Random Balance ensembles for multiclass imbalance learning. Knowledge-Based Systems 193, 105434.
[5] Díez-Pastor, J. F., et al. (2015). Diversity techniques improve the performance of the best imbalance learning ensembles. Information Sciences 325, 98–117.
[6] Kittler, J., et al. (2002). On combining classifiers. IEEE TPAMI 20(3), 226–239.
[7] Duin, R. P. W. (2002). The combining classifier: to train or not to train? Proc. ICPR Vol. 2, 765–770.
[8] Zhang, Chongsheng, et al. "A systematic review on long-tailed learning." IEEE Transactions on Neural Networks and Learning Systems (2025).

**Questions:**

Here are major points i would like a clarification in a rebuttal based on the weaknesses above:

- Your paper's central contribution is the "Bayesian Class-wise Trust-Weighted Ensemble," where the trust weight is defined as the class-wise precision. However, you also claim BESTI can "flexibly integrate" with external weighting schemes like LDAM or DRW. If these schemes modify the combination scheme, doesn't this alter the fundamental definition of your Bayesian trust weight, potentially breaking the theoretical justification? Could you clarify what "integration" means here and how the core Bayesian trust principle is preserved when using these external methods? Would it be that other components (Structured sampling) that can be coupled with other models?
- Could you provide a more complete ablation study that isolates the contribution of each core component? Specifically, what is the performance of BESTI without SS (e.g., using a baseline like Random Balance) but with your trust weighting, to show SS's unique benefit?
-  the method is stated "parallelizable," yet the algorithm and sampling seem sequential. Please clarify whether the training procedure can be done entirely in parallel.
- What is the cost-performance trade-off (e.g., a plot of accuracy vs. ensemble size/inference time) ?
-  The weighting mechanism uses class-wise precision. How is this fundamentally different from, or an advancement over, established "local accuracy" or "classifier competence" concepts from prior ensemble literature (see references above).
- How does sampling diversity (subset) correlate with ensemble diversity in model prediciton (see references above about diversity measures)?
- Lastly

---

### Official Review · Reviewer_dnnP · 2025-10-29

**Soundness:** 1
**Presentation:** 2
**Contribution:** 2
**Rating:** 2
**Confidence:** 5

**Summary:**

BESTI, an ensemble framework for long-tailed classification that (1) creates multiple sub-training sets via threshold-based structured sampling (progressively downsampling head classes), and (2) aggregates independently trained models with a Bayesian class-wise trust weighting, where “trust” is each model’s per-class precision estimated from a confusion matrix. Experiments on CIFAR-100-LT, ImageNet-LT, Places-LT claim consistent gains over a very small pool of outdated baselines (CE, Balanced Softmax, MiSLAS, LADE, RIDE, SADE). With older baselines, modest methodological novelty, limited metrics, and potential trust-weight leakage, the paper does not meet ICLR’s high standards on novelty and soundness in long-tail learning. I recommend reject.

**Strengths:**

+ The paper is clearly written and modular: structured clipping → independent training → precision-weighted ensembling; the method is easy to follow, understand, and reproduce.
+ Ablations across clipping schedules (exponential, uniform, quantile, k-means) and ensemble size k help demystify design choices; some runs include seeds/± values.
+ Results are reported on three standard LT benchmarks with many/med/few breakdowns and macro (“All”) accuracy.

**Weaknesses:**

+ One of my biggest gripes with this paper is the outdated baseline suite. The strongest comparison is to SADE (2022); others are RIDE (2020), MiSLAS (2021), LADE (2021), Balanced Softmax (2020)—all pre-2023. No recent long-tail advances (post-2022), no modern calibration-aware or distributionally-robust LT losses, and no contemporary ViT-era decoupling/tuning lines are included. With only older baselines, the competitiveness claim is weak for ICLR 2026.
+ BESTI is essentially cluster/threshold-guided undersampling + ensembling with precision-weighted voting; the “Bayesian” part reduces to the textbook identity that posterior(correct|pred=c) = precision (proved from the confusion matrix). This is a sensible heuristic, but not a conceptual leap beyond classic undersampling ensembles and class-wise weighting.
+ The paper computes each model’s confusion matrix on the training set (even if trained on a clipped subset) to derive precision weights. That is not a standard, leakage-aware protocol; it couples weights to training distribution quirks rather than a held-out set and can overstate “trust” for heads.
+ I find the evaluation scope too narrow for a claim of broad LT robustness. Metrics center on accuracy by shot bins; there is no calibration, tail-recall@k, AUCPR, or cost/threshold-aware analysis—important in long-tail. The paper also lacks open-set / class-presence shift probes and label-noise stress tests that are common in recent LT work.
+ BESTI trains k models; aside from a few ± numbers, there’s no wall-clock or energy comparison vs. strong single-model LT methods or lighter ensembles. The KL-based rule to stop at k is heuristic and only shown as a trend plot.
+ I  my opinion the current ablation study don’t isolate what truly matters. Gains could stem largely from downsampled re-training + ensembling. There’s no comparison to (i) simple uniform ensembling of k independently seeded models on the full data, (ii) head class temperature/threshold calibration, or (iii) precision weighting computed on a held-out split. The small drop from replacing trust with uniform averaging suggests the weighting has limited marginal effect.

**Questions:**

+ Why are post-2022 long-tail methods absent? Please add competitive 2023–2025 LT baselines and recent calibration/DRO objectives; otherwise the SOTA claim is not credible for ICLR.
+ Can you re-estimate class-wise precision on a held-out validation (or via cross-fitting) instead of the training set to avoid biasing the “trust” weights?
+ How does BESTI compare to a naïve k-model ensemble trained on the full data with different seeds (no clipping), and to simple per-class threshold calibration? Please provide matched-compute studies.
+ Please report calibration (ECE/ACE), AUCPR, tail-recall@k, and cost-sensitive metrics, and add label-noise and class-presence shift stress tests to reflect modern LT evaluation.
+ Your KL-based stopping rule is heuristic; can you give a compute/accuracy Pareto (k vs. hours/energy) and justify k in practical compute budgets?

---

### Official Review · Reviewer_CSPN · 2025-11-01

**Soundness:** 2
**Presentation:** 2
**Contribution:** 2
**Rating:** 2
**Confidence:** 3

**Summary:**

This paper proposes a multi-class classification algorithm to handle the issue of class imbalanced that combined structured sampling and Bayesian interpretation. The experimental results outperform some existing methods. However, the paper lacks rigor in its details and a detailed explanation of the issue solved by each module, limiting its overall originality and impact.

**Strengths:**

1. The paper handles the issue of class imbalance in multi-class classification, which is practically meaningful.
2. The proposed method was empirically validated, showing superior performance compared with several SOTA.

**Weaknesses:**

1. The connection between theoretical derivation and practical implementation is weak; some notations and their roles are ambiguous.
2. The description of the method part is not clear enough, some symbols lack definition, which makes it difficult to read, and the details are not rigorous enough.
3. The level of originality is limited, as the method largely combines existing ideas into a single framework without strong theoretical innovation.
4. The analysis, compare and discuss of recent related work are little.

**Questions:**

1. Please state key quantitative results (e.g., performance metrics) to underscore the effectiveness of the proposed method at the end of the abstract.
2. Enrich the related work by discussing more recent advancements in multi-label learning.
3. The latest method that the authors compare is in 2022, which is a bit old. The more recent algorithms should be compared.
4. C_{max} and r are two key parameters. How do they affect the performance of the algorithm?
5. Computational experience should be better explained. What's the computational burden of the proposed method?

---

### Comment · Area_Chair_LFK3 · 2025-11-27

Dear Authors and Reviewers,

The discussion phase will end soon. If you want to further discuss comments and replies with each other, please post your thoughts by adding official comments.

Thanks for your efforts and contributions to ICLR 2026.

Best regards,

Your Area Chair

---

### Meta-Review · Area_Chair_LsfE · 2026-01-05

**Summary:**

While the paper targets the relevant problem of long-tailed classification, the reviewers unanimously recommend rejection due to fundamental issues with novelty and rigor. The method is largely seen as a repackaging of existing resampling and ensemble techniques. The 'Bayesian trust' mechanism reduces to standard class-wise precision, offering little innovation over established competence-based approaches. The empirical study relies on outdated (pre-2023) baselines, single-run results, and narrow metrics. Critical assessments of calibration, robustness, and computational cost are missing. Key issues remain unresolved, including unclear theoretical grounding, potential training data leakage, insufficient ablations, and ambiguous claims regarding model diversity. The authors did not provide response to the reviewers' queries. Given the significant weaknesses in technical originality and modern validation, the work is not ready for publication.

**Reviewer Concerns:**

No rebuttal was submitted.

**Reviewer Scores:**

All reviewers recommended rejection.

---

### Decision · Program_Chairs · 2026-01-26

Reject